# Evolutionary selection of biofilm-mediated extended phenotypes in *Yersinia pestis* in response to a fluctuating environment

Yujun Cui[1,7], Boris V. Schmid[2,7], Hanli Cao[3], Xiang Dai[3], Zongmin Du[1], W. Ryan Easterday[2], Haihong Fang[1], Chenyi Guo[1], Shanqian Huang[4], Wanbing Liu[1], Zhizhen Qi[5], Yajun Song[1], Huaiyu Tian[4], Min Wang[1], Yarong Wu[1], Bing Xu[4], Chao Yang[1], Jing Yang[4], Xianwei Yang[1], Qingwen Zhang[5], Kjetill S. Jakobsen[2,8*], Yujiang Zhang[3,8*], Nils Chr. Stenseth[2,6,8*] & Ruifu Yang[1,8*]

*Yersinia pestis* is transmitted from fleas to rodents when the bacterium develops an extensive biofilm in the foregut of a flea, starving it into a feeding frenzy, or, alternatively, during a brief period directly after feeding on a bacteremic host. These two transmission modes are in a trade-off regulated by the amount of biofilm produced by the bacterium. Here by investigating 446 global isolated *Y. pestis* genomes, including 78 newly sequenced isolates sampled over 40 years from a plague focus in China, we provide evidence for strong selection pressures on the RNA polymerase ω-subunit encoding gene *rpoZ*. We demonstrate that *rpoZ* variants have an increased rate of biofilm production in vitro, and that they evolve in the ecosystem during colder and drier periods. Our results support the notion that the bacterium is constantly adapting—through extended phenotype changes in the fleas—in response to climate-driven changes in the niche.

[1] State Key Laboratory of Pathogen and Biosecurity, Beijing Institute of Microbiology and Epidemiology, Beijing 100071, China. [2] Centre for Ecological and Evolutionary Synthesis (CEES), Department of Biosciences, University of Oslo, Blindern N-0316 Oslo, Norway. [3] The Center for Disease Control and Prevention of Xinjiang Uygur Autonomous Region, Urumqi 830002, China. [4] State Key Laboratory of Remote Sensing Science, College of Global Change and Earth System Science, Beijing Normal University, Beijing 100875, China. [5] Key Laboratory for Plague Prevention and Control of Qinghai Province, Qinghai Institute for Endemic Diseases Prevention and Control, Xining 811602, China. [6] Ministry of Education Key Laboratory for Earth System Modeling, Department of Earth System Science, Tsinghua University, Beijing 100084, China. [7] These authors contributed equally: Yujun Cui, Boris V. Schmid. [8] These authors jointly supervised this work: Kjetill S. Jakobsen, Yujiang Zhang, Nils Chr. Stenseth, Ruifu Yang. *email: k.s.jakobsen@ibv.uio.no; xjsyzhang@163.com; n.c.stenseth@ibv.uio.no; ruifuyang@gmail.com

Plague is an endemic disease in many wildlife rodents across Asia's montane steppes, semi-arid deserts and mountain ecosystems[1,2]. In our current understanding, *Yersinia pestis*, the causative agent of plague, persists in these natural foci through an unbroken chain of transmission, where fleas become infected when sucking blood from plague-infected rodents and transfer the disease to other rodents during subsequent blood meals. Flea-borne transmission of plague has long been known to occur by two main modes. The first is biofilm-dependent transmission, where *Y. pestis* forms a partial or complete biofilm in the proventriculus (foregut) of the flea over a period of days to weeks that hampers, or stops new blood meals from reaching this organ. The flea, deprived of food and fluids, begins to feed more frantically, regurgitating infected blood in the process. The second transmission mode is known as early-phase transmission[3]. While the existence of the transmission route is well-documented, the mechanism of transmission is as of yet unknown. Early-phase transmission in fleas fed on mice blood has been shown to transmit in similar efficiency in biofilm-deficient *Y. pestis* strains as in biofilm-producing strains[4]. Both modes of transmission are extensively reviewed by Hinnebusch et al.[5].

The two transmission modes are in a trade-off with each other. Increased levels of biofilm formation lead to a better ability of the bacterium to maintain itself in the foregut of the flea[6], and thus improve its chances to persist in the flea gut for long enough to successfully transmit through blockage-induced transmission. However, these increased levels of biofilm formation decreases the efficiency of early-phase transmission[4]. This trade-off between blockage-induced transmission and early-phase transmission occurs in the domain of normal-to-high levels of biofilm production. In the domain of normal-to-reduced levels of biofilm production, the relationship between blockage-induced transmission and early-phase transmission becomes more complex, depending on the host blood source. For mice, normal levels of biofilm formation, or even the complete absence of biofilm formation do not seem to negatively impact early-phase transmission[4], while they do negatively impact blockage-induced transmission. In rats and guinea pigs, low or absent levels of biofilm formation also negatively impact early-phase transmission, as some degree of biofilm formation appears to be involved in an interaction between host blood and *Y. pestis* that drastically boosts the efficiency of early-phase transmission[7]. In addition, at reduced levels of biofilm production, more fleas will be at a stage of partial, rather than complete blockage of the proventriculus. As partially blocked fleas can still hydrate themselves, yet also spread plague, reduced levels of biofilm production increase the bacterium's ability to survive in hibernating fleas[8], and extends the lifespan of infected fleas[5]—both factors improving the persistence of plague in the ecosystem. Outside of the flea vector, biofilm formation may play a role in rodent hosts, possibly as an early defense against the innate immune system[9,10], and could play an as-of-yet unknown role for plague in the hypothetical soil compartment[11,12].

The rate of biofilm production is a genetically determined trait of *Y. pestis*[5], and alters the feeding behavior of the flea by blocking its proventriculus. Among bacteria, the ability to modulate the fleas' feeding behavior appears to be a feature specific to *Y. pestis*[13]. As such, this transmission trade-off is an example of variation in an extended phenotype trait (as originally introduced by Dawkins[14]) of the bacterium expressed in the flea (its vector). According to the extended phenotype perspective, the genotype of the bacterium influences the phenotype and/or the behavior of the host (flea) species through the type of biofilm formation affecting the flea's "feeding" behavior).

To study whether evolutionary trade-offs like the one above occur in *Y. pestis* in a natural setting, we investigated the temporal dynamics of *Y. pestis* within a wildlife plague reservoir of ground squirrels (*Spermophilus undulatus*) and their fleas (*Citellophilus tesquorum*). The Guertu plague reservoir is located in the Tien Shan mountains in north–west China. It exhibits strong seasonality and large interannual variation in temperature and precipitation, large fluctuations in the population density of ground squirrels and their fleas, as well as in the prevalence of *Y. pestis* (Supplementary Fig. 1, Supplementary Data 1). We hypothesized that these large fluctuations in the environment might exert sufficient selection pressure on the bacterium for such signals to be visible in a longitudinal time series of bacterial genome.

To trace the associations between evolution of *Y. pestis* and the ecological factors, we collected *Y. pestis* isolates and climate information over 40 years from a single natural plague focus. Comparing the *Y. pestis* genomes, we find a particularly strong selection signal in the RNA polymerase ω-subunit encoding gene *rpoZ*. In vitro, these *rpoZ* variants display an overdeveloped biofilm-producing capability, which suggests that these variants could have an altered extended phenotype in terms of flea transmission, following the trade-off between blockage-induced transmission and early-phase transmission of the disease. Using the phylogenetic tree of *Y. pestis* in the Guertu plague reservoir to estimate when these *rpoZ* variants likely evolved, we link their occurrence to periods of dry and cold weather. The link between climate, flea behavior, and genomic changes in *Y. pestis* opens up new venues for exploring the association between the bacterial evolution of plague and the way it spreads through natural foci and beyond.

## Results

**Low genetic diversity present in natural isolates of plague.** Between 1967 and 2006, 78 *Y. pestis* isolates were collected from the Guertu natural plague reservoir (Fig. 1a), and their genomes were sequenced in 2013 (Supplementary Data 2). Of these isolates, 71 had a highly consistent genome size, but seven isolates had lost the putative pathogenicity island *GI03* or the putative pathogenicity island *pgm locus*, and one lost its pMT plasmid, which we marked as the accessory genome (Supplementary Fig. 2). In the shared core genome of 4.3 million base pairs, we found a total of 54 single-nucleotide polymorphisms (Supplementary Data 3) and 76 insertions or deletions (Supplementary Data 4). A phylogenetic tree of the isolates constructed based on the 54 SNPs using a Bayesian relaxed-clock model in BEAST2[15] shows that between 1967 and 2006 multiple lineages of *Y. pestis* existed in parallel within the ecosystem, without any introductions of new strains from outside (Fig. 1b).

**Positive selection in a biofilm-associated gene.** In order to assess whether there were signs of positive selection pressure within these 78 plague isolates, we screened for clusters of variations that occurred in such small regions of the circular chromosome that they had a low probability of occurring under a neutral substitution model, in which variations are assumed to be randomly distributed across the genome[16] (Table 1). The most significant cluster ($P < 0.00001$ by permutation testing, see the Methods section and the code repository for more details) was a cluster of eight independent variations, including three SNPs and five indels, in the small RNA polymerase ω-subunit *rpoZ* gene (276 bp). As most strains experienced less than five passages before being kept as freeze-dried powder, we strongly suspect that the observed clustering of variations in the *rpoZ* gene is not an artifact of laboratory passages of the bacterium, but is caused by selection pressure from natural plague ecosystem. The *rpoZ* gene is known to alter colony morphology in both *Streptomyces kasugaensis* and

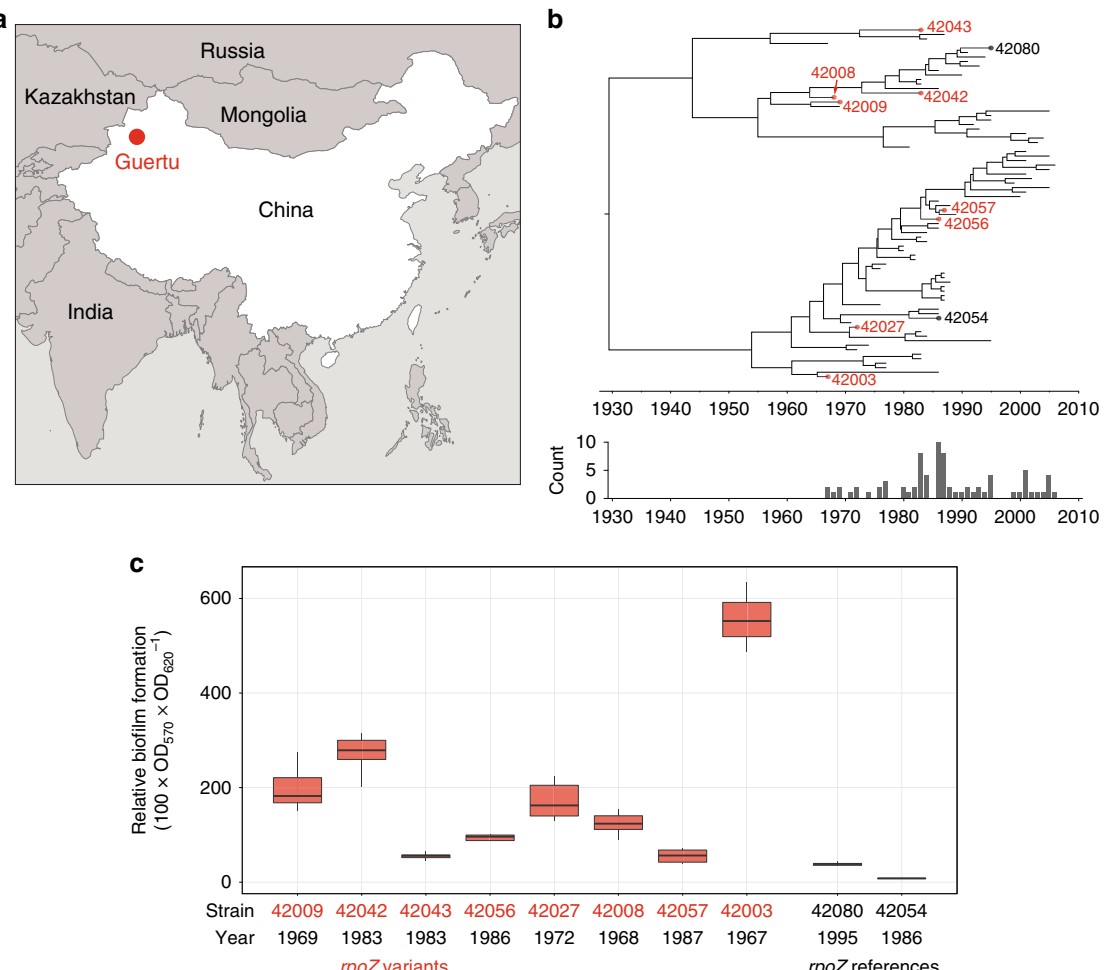

**Fig. 1 The geographical position, phylogeny, and biofilm-formation capability of *rpoZ* variants. a** The geographical position of the Guertu plague foci sampling site. The map was created based on the public geographical data downloaded from OpenStreetMap (http://openstreetmap.org) and was licensed under the CC BY-SA (https://www.openstreetmap.org/copyright). **b** Maximum clade credibility phylogeny of 78 *Y. pestis* strains in Guertu, based on 54 SNPs. Red dots indicate the eight isolates with *rpoZ* mutations, black dots the two *rpoZ* wild-type isolates used as control. The labels are the strain ID number. A full version of the tree is provided in Supplementary Fig. 5. The plot below the tree shows the sampling distribution over the years. **c** The in vitro biofilm-formation capacity of the *rpoZ* mutants (fivefold repeated experiment) and the control strains, as measured by crystal violet staining. Boxplots depict the upper, median, and lower quartiles, individual dots indicate outliers that lie outside of 1.5 times the interquartile range, and vertical lines indicate the range of all values except for outliers. The Source data of Fig. 1c are provided as a Source Data file.

**Table 1 Clusters of variations across 78 *Y. pestis* isolates†.**

| K* | N# (in bp) | CO92 position | Probability of cluster under neutral substitution model | Genes or genome regions involved |
|---|---|---|---|---|
| 8 | 127 | 52,546-52,672 | $P < 0.00001$ | *rpoZ* (RNA polymerase ω-subunit) |
| 5 | 2904 | 2,326,351-2,329,254 | $P = 0.00013$ | *CutC* (copper homeostasis protein); YPO2050; intergenic; YPO2051 |
| 2 | 2 | 358,874-358,875 | $P = 0.0034$ | *aspA* (aspartate ammonia-lyase) |
| 2 | 5 | 581,519-581,523 | $P = 0.013$ | Intergenic; YPO0535–YPO0536 |
| 2 | 15 | 3,705,090-3,705,104 | $P = 0.048$ | YPO3321; YPO3322 |

†Two small clusters (at CO92 position 2,577,933 and 2,606,316) were excluded as they were the result of copy-number variations in tandem repeat loci
*K, the number of variations within a cluster
#N, region size of the cluster

*Mycobacterium smegmatis*[17,18], and deletion of the *rpoZ* gene lowered biofilm formation in *M. smegmatis*[18]. Such a selection pressure on the *rpoZ* gene in the Guertu plague ecosystem is of interest with regard to the trade-off between biofilm-induced blockage and early-phase transmission of *Y. pestis*.

**rpoZ variants are found worldwide**. Notably, the high frequency of polymorphisms in the *rpoZ* gene is not confined to the Guertu strains. By comparing 368 public available genome sequences of *Y. pestis* strains from across the globe (Supplementary Data 5), we found three additional variations in the *rpoZ* gene (one SNP and

two indels) distributed across 35 *Y. pestis* strains, most of which occurred in the 0.PE phylogroup in the Former Soviet Union (FSU) countries (Supplementary Fig. 3, Supplementary Data 6). Interestingly, two FSU strains carried the exact same variation as one of the Guertu *rpoZ* variants (a single-nucleotide deletion at 52672 at position in CO92 genome, Supplementary Fig. 4). This may indicate a signal of convergent evolution (including selection for independent de novo mutations or selection for standing variation) and further supports that the *rpoZ* gene is under strong selection.

In the Guertu ecosystem, none of the *rpoZ* variations were found in more than one sample, and no *rpoZ* variations were found post 1987. In part, the uneven sampling distribution plays a role here (lower panel, Fig. 1b), with just 29 samples collected after 1987. Given the relatively low odds of finding *rpoZ* variants in general (8/78 samples in Guertu, 35/368 samples across sequenced *Y. pestis* strains), not finding any *rpoZ* variants in the 29 samples collected post 1987 are low, but not highly unlikely $\left[ \left(1 - \frac{8}{78}\right)^{29} = 0.043 \right]$. Furthermore, those low odds depend on the assumption that *rpoZ* variants are equally likely to be selected for in every year, which we show in the climate analysis below not to be the case.

**Biofilm production rates vary among natural isolates**. Comparing the eight *rpoZ* variants and two *rpoZ* references (strains that have an *rpoZ* sequence identical to the CO92 reference strain) shows that the former had a significantly higher level of biofilm formation after 24 h, as measured by crystal violet staining (ANOVA test, $F = 161.48$, $P < 0.001$) (Fig. 1c). Furthermore, the ten tested isolates showed a large amount of variation in biofilm production rate between them, despite their small genetic differences in just 25 loci across the whole genome (Supplementary Table 1). Many of these 25 loci appear to be involved in processes known to affect biofilm formation (Supplementary Table 1).

***rpoZ* variants correlate with colder and drier climates**. A general caveat when comparing biofilm-formation rates in vitro is that effect sizes can be different from those in vivo[19]. However, we do find additional support for a phenotypic effect of the *rpoZ* variants in the distinct ecological circumstances during which

these variants were selected for. As none of the *rpoZ* variations was observed in more than one sample, each *rpoZ* variation can be described as being selected for somewhere in the period between the phylogenetically estimated date when the *rpoZ* variant sample branched off from the main tree and the sampling date of the variant (Fig. 1b). We statistically compare the average local climate conditions during the period in which the *rpoZ* variants developed with the same period calculated for the *rpoZ* references. We extend the comparison period for all samples by ±1.3 years to account for uncertainty in the sampling date (samples were collected between June and August), to allow for a year of trophic cascade effects prior to the branching off to play a potential effect, and to round the duration up to full months—see the Methods section and the code repository for more details. We find that the phylogenetic branches that contained the *rpoZ* variants existed during periods in which the average monthly temperature and average monthly precipitation levels were lower than that were typical for the *rpoZ* references (Fig. 2a). The statistical significance of a correlation between climate and *rpoZ* variants was tested for the colder and drier climate hypothesis and seven other possible climate hypothesis (colder, warmer, wetter, or drier climate, or the four possible combinations thereof). Statistical significance was estimated through permutation testing by counting how frequently the climate of a random set of eight *rpoZ* references (sampled without return out of the 70 *rpoZ* references) was more extreme than that of the *rpoZ* variants (more extreme here means that if we are testing for colder and drier climate, the random sample would have to be both colder and drier). We found the climate in the Guertu ecosystem to be significantly more often colder and drier ($P$-value < 0.0092, permutation test) during the time periods in which the phylogenetic branches existed in which *rpoZ* variations occurred, than was the case for the *rpoZ* references. The other seven climate hypotheses were borderline significant or nonsignificant, with the two most significant of these (at $P$-values of $P < 0.0396$ and $P < 0.0304$, permutation tests) indicating that *rpoZ* variants arose during climate periods that were drier, and warmer and drier, respectively. While neither of these hypotheses would remain significant after correcting for multiple testing, they may indicate that a lack of precipitation is the more important factor in the rise of *rpoZ* variants, and that the role of temperature is less certain.

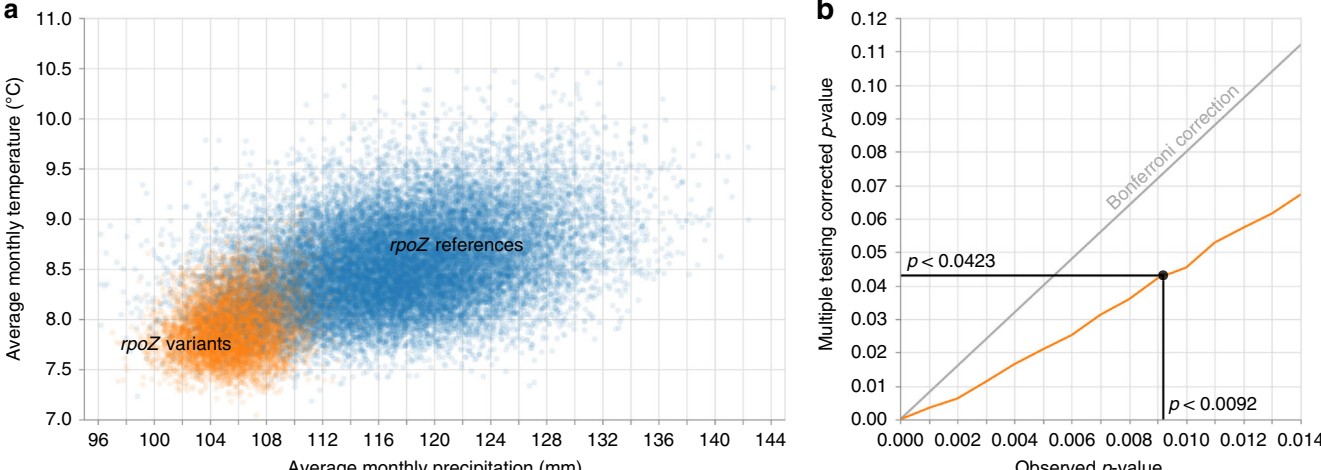

**Fig. 2 The relationship between climate conditions and appearance of *rpoZ* variants. a** average climate conditions during the periods in which the *rpoZ* variants were generated (as per the branch lengths of the phylogenetic tree, plus a year prior to account for cascading effects of climate on hosts and vectors to plague), compared with the climate conditions for *rpoZ* references. **b** correcting the observed *P*-value of <0.0092 for the pattern of low average monthly precipitation and temperature observed in panel **a** for multiple testing of eight related climate hypothesis, using permutation testing (orange line) as opposed to the Bonferroni correction (which assumes unrelated tests). The corrected *P*-value is *P* < 0.046.

Applying a Bonferroni correction to correct for multiple testing of the different climate hypotheses would be overly stringent here, as our eight hypotheses are correlated with each other. We therefore corrected for multiple testing by using another layer of permutation testing, in which we repeatedly marked eight randomly selected samples (without return) as our "samples of interest" (like the eight *rpoZ* variants in our main result), ran the earlier described permutation test for the eight climate hypotheses again. We then scored how often we would find significant correlations between these randomly generated "samples of interest" and one of the eight climate hypothesis that had a lower P-value than the P-value < 0.0092, we found when using the actual *rpoZ* variants as our sample of interest (Fig. 2b). The frequency with which we find that these randomly generated "samples of interest" result in P-values < 0.0092 is then our multiple test-corrected P-value. After correcting, we estimated the P-value for the climate signal of colder and drier years correlated to the rise of *rpoZ* variants to be P < 0.0423 (estimated by permutation testing). For more details on the permutation testing, see the Methods section and the code repository.

## Discussion

How exactly colder and drier years affect the different trophic levels in the Guertu plague ecosystem is hampered by the limited amount of surveillance data available (Supplementary Fig. 1, Supplementary Data 1). From other studied plague ecosystems, located in the Prebalkhash desert of Kazakhstan, in Inner Mongolia in China and in Montana and Colorado in the USA[20–22], we know that the strength of the trophic cascade relationships between climate and plague can differ between ecosystems, being evident in some, and absent in other plague ecosystems. For Colorado, no relationships with climate were found[23], and for the Mongolian gerbil in Inner Mongolia, the cascade effects were complicated and the overall effect of increases in temperature and precipitation on plague prevalence not reported. For Kazakhstan and Montana, we see positive effects of increased precipitation, as well as positive effects of increased temperatures (up to a limit) for plague prevalence[20,21,23]. The studies in Kazakhstan further elucidate the relationship between climate and plague prevalence, linking it to an increase in both the flea burden per rodent[20] and the overall rodent density[21], which in turn facilitates the spread of plague[24]. Extrapolating from these findings, the relationship we report here between a colder and drier average climate in Guertu and the evolution of *rpoZ* variants could indeed relate to depressed rodent and flea abundances during the period in which *rpoZ* the selected variants persisted in the ecosystem, up to the date of their sampling.

Using a combination of sequencing, surveillance, and ecological information, we present a rare look into the evolutionary dynamics of *Y. pestis* within a natural plague reservoir. We find continuous and substantial genetic variation, similar to that reported in the urban reservoir of plague below Mahajanga, Madagascar[25,26]. The observed continuous occurrence of genetic variation in both Guertu and Mahajanga is in sharp contrast to the unresolved puzzle of the mostly clonal expansion of *Y. pestis* through millions of medieval Europeans during the Black Death pandemic[27,28].

In the Guertu natural plague focus, eight independently evolved *rpoZ* variants appear to have been selected during colder and drier periods in the ecosystem. The limited ecosystem surveillance data and overlap thereof with the time frame during which the *rpoZ* variants likely evolved, make it difficult to test for causal links between the different trophic layers. However, based on numerous studies on the flea-borne transmission of plague, reviewed in Hinnebusch et al.[5], a causal link between biofilm-

formation capability and the trade-off between biofilm-induced and early-phase flea-borne transmission of *Y. pestis* seems plausible. In this respect, the extended phenotype triggered by the full blockage of the foregut of the fleas is one of frenzied feeding and regurgitation of the bacteria, and is estimated to increase the probability of transmission of the disease to 25–50% per flea per bite, compared with the low 0–10% probability of early-phase transmission per flea during the 3-day time window of early-phase transmission[29]. The probability of plague transmission is ultimately dependent on multiple factors, such as flea species, temperature, and blood source[7,30,31].

Theoretical explorations of the conditions under which blockage-induced transmission is favored over early-phase transmission are still in their early stages, but do suggest that both transmission strategies can persist simultaneously in a fluctuating environment, and become more or less dominant, due to genetic selection based on the conditions that favor their spread[32]. A possible hypothesis, in need of further (experimental) investigation, is therefore that the selection for increased biofilm production through variations in the *rpoZ* gene is caused by a trophic cascade, where cold and dry climate negatively affects rodent and flea densities, which in turn favors biofilm-induced plague transmission over early-phase transmission. Although there is strong experimental evidence for the importance of biofilm production in the flea-borne transmission of plague, we cannot exclude that the selection of the *rpoZ* variants in the Guertu ecosystem may be related to other factors, such as the poorly understood ability of the bacterium to survive in the soil[11,12], or the role that the expression of biofilm-related proteins plays in the rodent host[9,10].

The ability of *Y. pestis* to regulate its preferred path of flea-borne transmission through an extended phenotype linked to biofilm production is a remarkable adaptation to its local environment—essentially through an environmentally/climatically driven response. The consequence of this adaptation stretches beyond the local flea species when plague spillovers reach ecto-parasites associated with humans, domestic or peri-domestic animals, and alters their vector efficiency, for better or worse.

## Methods

**Characteristics of the surveillance**. Plague-surveillance work in the Guertu region began in 1964. Routine surveillance, including the type and distribution of host/vector, host animal density, flea index, and bacteria isolation was performed each year, and the first *Y. pestis* strain in Guertu was isolated in 1967. The sentinel site in Guertu was established in 1983. Plague-surveillance staff lived and worked at the sentinel site from May to October each year, when *Spermophilus undulatus* and *Marmota baibacina*, the hosts of *Y. pestis*, emerge from hibernation. Once the sentinel site was established, seropositivity rate was also measured. Host density, flea index, seropositivity rate, and bacteria isolation were determined by standard protocol described in the National Scheme of Plague Surveillance released by the National Health and Family Planning Commission of the People's Republic of China.

Plague-surveillance data are kept by the Xinjiang CDC, and some of it has been published in previous reports[33–38]. This study used data from 1982 to 2010 (Supplementary Fig. 1, Supplementary Data 1). Seventy-eight Guertu *Y. pestis* strains were stored and provided to us by the Chinese Medical Bacteria Center of Management and Preservation in Xining, China (Supplementary Data 2).

**Host density**. Quadrat sampling was used to determine ground squirrel density of Guertu plague focus since 1970s until 2017[39]. Each quadrat had an area of 10,000 m². Five to ten quadrats were randomly selected from the Guertu region each year, with a distance of greater than two kilometers between each quadrat. For each quadrat, the number of ground squirrels was counted 1 h after sunrise by telescope. The count lasted for 2 h, and was repeated for 2 consecutive days. The highest count recorded within the 2 days was used to calculate host density by dividing the number of ground squirrels by the quadrat area. Host density for the entire Guertu region was calculated as the average density between quadrats. The counts were performed twice each year (in May and July, respectively), and the average value of the two measurements denoted the annual host density.

**Capture of host animals**. The host animals were captured in region that excluded the quadrats using for determining the host density. The mouse traps were set near the burrows of host animal, and in every half an hour the traps were checked. The captured animal was executed in the field, and the body was sealed in a hermetic bag to be carried back to laboratory for counting fleas, collecting serum, and organ samples for further investigation.

**Flea index**. The bodies of captured animals in the hermetic bags were fumigated by using diethyl ether. For each captured animal, the number of fleas was counted and recorded independently. The flea index of the year was calculated as the total number of collected fleas divided by the total number of captured animals in the year.

**Bacteria isolation**. For each captured animal, samples of lymph nodes, blood, heart, liver, spleen, kidney, and lungs were divided into two portions. One portion was used to inoculate onto selective agar plates (BIN agar, a brain heart infusion agar base with the added selective agents of irgasan and cholate salts)[40]. The cultures were observed for 5 days, and were discarded if no suspected colonies of *Y. pestis* were observed. Suspected colonies of *Y. pestis* were transferred onto LB plates to ensure a pure culture. The other portion of organs of every tenth ground squirrel were fully mixed and injected intraperitoneally into a healthy mouse in the laboratory to observe whether this resulted in a *Y. pestis* infection. This was done to reduce the false negative rate of *Y. pestis* isolation, caused by low bacterial load within the natural hosts and vectors. The mice were observed for 7 days after inoculation. If disease symptoms of *Y. pestis* infection (e.g., fever, ruffled fur, and dysthymia) were observed, the samples of lymph nodes, blood, heart, liver, spleen, kidney, and lungs were inoculated onto selective agar plates for bacterial isolation. Similarly, the fleas collected in the captured animals were also divided into two portions. One was washed with physiological saline, ground to emulsion, and then inoculated onto the selective agar plates. Another was also ground to emulsion and then injected intraperitoneally into mouse (20 fleas of the same species per mouse), after which the culturing and bacterial isolation of plague from flea samples followed the same protocol as that of the animal organ samples. Identification of *Y. pestis* was done by microscopic examination and bacteriophage lysis test[39,41]. Isolates identified as *Y. pestis* were sent to the Chinese Medical Bacteria Center of Management and Preservation in Xining, China, and stored as freeze-dried powder. Most of the isolates experienced less than five passages, and then were kept as freeze-dried powder. Two strains, 42005 and 42010, had been passaged for 40 and 31 times, respectively, before being sent to the culture center.

**Sequencing and assembly**. Paired-end libraries with an insert size of 500 bp were constructed for each Guertu isolate, and whole-genome sequencing was done by Illumina HiSeq 2000 (Illumina Inc. USA) following the manufacturer's instructions. The read length was 90 bp, and >500 Mb raw data were generated for each genome. The paired-end sequencing reads were assembled de novo using SOAP-denovo[42] (Supplementary Data 2).

**Variation detection**. SNPs were identified by aligning each assembled Guertu genome against a reference genome, CO92 (accession number: NC_003143.1) using MUMmer software[43]. SNPs located in repetitive regions, or supported by less than ten high-quality paired-end reads (quality score > 20), were removed[44]. In total, 54 high-quality SNPs (28 non-synonymous, 10 synonymous, and 16 intergenic) were identified in 78 Guertu genomes (Supplementary Data 3). We selected one Guertu genome, 42003, as a reference genome and repeated the SNP calling process, but no new SNPs were identified. The ancestral state for all SNPs was defined as the nucleotides in the allele of *Y. pseudotuberculosis* IP32953, and all alternative nucleotides were defined as derived state.

The Indels shorter than 50 bp were identified according to Cui et al.[45]. We aligned the Guertu genomes to the selected reference genome, 42003, by using LASTZ software[46] (http://www.bx.psu.edu/~rsharris/lastz/). Then the Indels were validated by mapping the reads of Guertu isolates separately to their assemblies, using BWA, and visualized by samtools[47]. All identified Indels were manually checked. The identified alignment errors and the Indels located at the end of contigs (<20 bp) were excluded from the final data set (Supplementary Data 4). The Indel loci and sequences of 300 bp surrounding the Indel were extracted to map against the CO92 genome to acquire their alleles, and the annotation information was obtained based on the position of alleles. Most Indels revealed double states. The state that equated to the nucleotide sequence in the allele of the CO92 genome was recorded as state "0", and the second state was recorded as state "1". Two Indels, ID0038 and ID0040, revealed three different states, and the third state was recorded as state "2". Both triplicate Indels were located at tandem repeat regions, and their state change was caused by a variation in the copy number of motifs.

We identified the core genome (common shared sequences of all 78 Guertu genomes, 4.31 Mb) and accessory genome (sequences that were absent in at least one genome, 0.29 Mb) for Guertu isolates, according to protocol from Cui et al.[44]. We mapped the accessory genome sequences against the reference CO92 genome to find their homologous and corresponding functional annotations. All fragments of the accessory genome were present in most of the Guertu genomes, i.e., fragment absence was only observed in a few of the Guertu isolates. Following the principle

of parsimony, we concluded that there were likely no gene gain events, and gene loss events occurred in only a few isolates. Interestingly, all gene loss events occurred on two known genome islands in the *Y. pestis* chromosome (Supplementary Fig. 2), with the exception of one strain, 42069, which lost its pMT plasmid. The first genome island is the *pgm* locus and its flanking sequences (YPO1884-1999). The *pgm* locus is a genome island associated with transmission in fleas and virulence toward the rodent hosts of *Y. pestis*[48,49], and its loss in *Y. pestis* genome has been observed in both natural isolates and experimental passage[3,49]. The second genome island is located at a putative pathogenicity island (GI03, YPO0602–642), encoding for adhesin, autotransporter, and protein kinase[50].

**Identification of variation clusters**. We searched for any cluster of variations on the circular main chromosome of *Y. pestis* that are located in a smaller stretch of base pairs than would be expected to occur in a frequency of <5% of the time anywhere in a genome where 129 variations are randomly distributed across the 4653728 base pairs. The above approach is described by Zhou et al.[16], who consider "the [binomial] probability that $k$ or more variations fall exactly within $n$ residues", and argue that a small probability would be a sign of positive selection.

In our implementation (of which the annotated source code and data files are available in the code repository), the significance of a cluster of $k$ variations within a span of $n$ base pairs is determined with a permutation test, in which we repeatedly ($\times 100,000$) generate fictional chromosomes in which 129 variations are randomly distributed across 4653728 base pairs, and count how frequently these fictional chromosomes contain at least 1 cluster of $k$ variations within a span of $\leq n$ base pairs. We further limit our search for clusters to those of size $k = [2,10]$, and $n \leq 10,000$, and manually exclude from the results any clusters that consist of a subset of variations of a significant larger cluster (where significant is defined as a $p < 0.05$ significance level). In addition, we excluded two small clusters (at CO92 position 2,577,933 and 2,606,316), as they are the result of copy-number variations in tandem repeat loci. Such loci have much higher mutation rates than other parts of the chromosome[51], and were therefore not investigated further. All significant clusters are shown in Table 1.

As *Y. pestis* has a circular chromosome, when mapping that chromosome to a linear sequence one might split up a cluster. However, the Guertu isolates conveniently have their first variation at position 52,546 when mapping to *Y. pestis* CO92. At the maximum window size we considered of $N = 10,000$ bp, there was no cluster that we could have missed that straddled the edges of the linearized sequence.

**Phylogeny reconstruction**. A total of 446 genomes were used in the phylogeny reconstruction, including 78 genomes of Guertu isolates and 368 global isolates downloaded from NCBI. The accession numbers of all genomes used are listed in Supplementary Data 5. We recalled SNPs for the 446 genomes using the same pipeline described above. The maximum likelihood tree (MLtree) was built based on concatenated SNPs with HKY model by using PhyML[52] with 100 bootstrap replicates, and visualized by the online tool iTOL[53]. The MLtree of 78 Guertu isolates and 368 public *Y. pestis* strains indicated that all Guertu isolates belong to a same phylogroup, 0.ANT1 (Supplementary Fig. 3), which is only distributed in the Guertu region according to previous studies[44,54].

We also built the maximum clade credibility tree for 78 Guertu isolates (Fig. 1; Supplementary Fig. 5) using the BEAST2 software package[15]. We evaluated the combinations of two clock models (strict clock and relaxed log-normal clock), two substitution models (HKY and GTR), and two population size change models (constant size and exponential population size), by Bayes factor test according to Cui et al.[44]. The log-normal clock + GTR model + exponential population size fit the data best, and were used to estimate the results. We performed three independent runs with 30 million MCMC iterations. The output of the three runs was combined, and the first 10% of chains were discarded as the burn-in phase using the program Tracer in BEAST2. The final effective sample sizes of all inferred parameters were above 200.

**Determination of biofilm production of *Y. pestis***. In the LB broth, bacterial cultures incubated overnight with shaking were diluted to $OD_{620} = 0.02$. Diluted cultures were added to a 24-well polystyrene plate, which were incubated with shaking at 230 rpm at 26 °C for 24 h, and the bacterial cell density was determined by measuring $OD_{620}$. After cells were removed, the wells were washed three times with distilled water, and the biofilm was stained with a 0.1% crystal violet solution (3 ml) at room temperature for 15 min, after which the wells were washed three times with distilled water. Crystal violet stain was solubilized in dimethyl sulfoxide (3 ml), and the absorbance at 570 nm ($OD_{570}$) was measured. The relative biofilm formation was expressed as $100 \times OD_{570} \times OD_{620}^{-1}$ (Fig. 1c).

**Collection of local temperature and precipitation data**. Local climatic data on precipitation and temperature at the weather station nearest to Guertu (station number 51334) were obtained for the study period from China Meteorological Data Sharing Service System (Supplementary Data 1).

**Statistics of the ecological analysis.** Among all 446 *Y. pestis* genomes analyzed in this study, the sequence of *rpoZ* gene is fully identical in 403 genomes. Here we defined the strains that carried major allele type of the *rpoZ* as "*rpoZ* reference" (identical to the CO92 reference strains). The other 43 genomes that carried minor allele types of *rpoZ* were defined as "*rpoZ* variants". As described in the article, we used permutation testing (i.e., resampling without returns, also known as re-randomization testing) to gain insight into the question whether the climate in Guertu was different (that is colder, warmer, wetter, or drier climate, or the four possible combination thereof) during the estimated time periods since the phylogenetic branches that contained the *rpoZ* variants split off from the main tree, compared with the climate in the time periods preceding the *rpoZ* references. We extended the actual climate period to compare between samples to be somewhat longer than the branch length indicated by the phylogenetic tree. We did so because the exact sampling dates are not known for the samples (we know sampling occurred in June–July–August), and we did not want to accidentally exclude months that might have been relevant for the selection for *rpoZ* variants. So, all samples were treated as having been sampled on the 1st of September, and thus had three additional months added to the duration indicated by the branch length to cover the sampling period. We also rounded up the number of duration of the periods to include the whole month (e.g., if the branch length indicated to consider the climate back to the 25th of June 1979, we would include the whole of June). Finally, we added a full year to the duration period considered, to allow for trophic cascade effects, where the climate could exert its selection pressure through affecting the conditions for rodent and or flea populations in a way that would take time to express themselves in changes in rodent and flea densities[22]. On average, thus the duration of the climate considered for each sample was the branch length indicated by the phylogenetic tree (on average 4.9 years) + 3 months + 0.5 month + 1 year, which when added together is on average 6 years, 2 months, and 10 days, counting back from the 1st of September.

The date of diverge for the internal nodes (and thus of the branch lengths between the main tree and the sampling date of the *rpoZ* variants and the *rpoZ* references) in trees generated by BEAST comes with a substantial amount of uncertainty. To capture some of this uncertainty in our analysis, we used the logfile of phylogenetic trees generated by the MCMC process of BEAST, under the same settings as were used in a later iteration of the phylogenetic analysis that used BEAST2 (i.e., statistically the same output). This logfile gave us 8000 potential phylogenetic trees to work (the logfile held 10,000 trees collected during the MCMC, each 6000 iterations apart. We dropped the first 20%). In each of these trees, we calculated the average temperature and precipitation (on a monthly resolution) for the eight *rpoZ* variants combined and for eight randomly selected (without resampling) *rpoZ* reference samples combined, and recorded how the randomly selected set compared with the eight *rpoZ* variant samples in terms of precipitation, temperature, or combinations thereof. We iterated 300 times over all 8000 trees, resulting in a permutation test with a total of 2.4 million iterations. In total, we tested eight climate hypothesis: *rpoZ* variants arise during warmer, colder, drier, or wetter years, or the four possible combinations thereof (warmer and wetter, warmer, and drier, etc), compared with the *rpoZ* references.

Because our eight climate hypotheses are partially overlapping with each other, a Bonferroni correction would be overly stringent. As indicated in the article, we corrected for multiple testing with another layer of permutation testing, where we repeatedly marked eight randomly selected samples (without resampling) as our "samples of interest", ran the permutation tests as described above again where we compared the eight selected samples against eight randomly selected samples from the "not samples of interest" pool, and scored how often we would have found by chance an equal of stronger *P*-value for at least one of the eight climate hypothesis (Fig. 2b). For each of 4000 iterations, we picked a new random set of eight "samples of interest", and for each of these iterations we ran the permutation test described above for 24,000 iterations. The code used is available in the code repository.

**Reporting summary.** Further information on research design is available in the Nature Research Reporting Summary linked to this article.

## Data availability

The genome sequences reported in this paper have been deposited in the National Center for Biotechnology (NCBI) database (www.ncbi.nlm.nih.gov) under BioProject PRJNA412676 [https://www.ncbi.nlm.nih.gov/bioproject/412676/]; individual accession numbers are listed in Supplementary Data 2). The source data underlying Fig. 1c are provided as a Source Data file. All other data are available in the Supplementary Data 1–6.

## Code availability

The BEAST XML file, as well as the code and data used to identify the clusters of mutations and for the climate analysis is provided in the code repository at https://doi.org/10.5281/zenodo.3279956.

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

## Acknowledgements

We appreciate all three generations of Xinjiang CDC employees for their >50 years of dedicated plague-surveillance work. We thank Joe Hinnebusch, Amine Namouchi, Jukka Corander, and John Michael Koomey for helpful discussions. This work is supported by the National Natural Science Foundation of China (No. 31430006), National Key Research & Development Program of China (2016YFC1200100), funding from the State Key Laboratory of Pathogen and Biosecurity (No. SKLPBS1405), Beijing Advanced Innovation Program for Land Surface Science; Beijing Natural Science Foundation (JQ18025); Young Elite Scientist Sponsorship Program by CAST (YESS) (2018QNRC001); the European Research Council (ERC) under the FP7- IDEAS550 ERC Program (Grant 324249), the Research Council of Norway (FRIMEDBIO project 288551), the University of Oslo, MLS grant #152950, and the Centre for Ecological and Evolutionary Synthesis (CEES) funded through the Research Council of Norway.

## Author contributions

R.Y., N.C.S., K.S.J. and Y.C. designed the study and coordinated the project. Specifically, R.Y. designed the study and the collection of the data; N.C.S. developed the theoretical framework for interpreting the data; H.C., X.D. and Y.Z. contributed strains and surveillance information for analysis; Y.C., B.V.S., C.G., S.H., K.S.J., H.T., Y.W., C.Y., J.Y. and X.Y. analyzed the data; Z.D., H.F., W.L., Z.Q., M.W. and Q.Z. performed the experiments; W.R.E., Y.S., and B.X. provided insightful comments, B.V.S., Y.C., K.S.J., N.C.S. and R.Y. wrote the paper. All authors approved the final version of the paper.

## Competing interests

The authors declare no competing interests.
