## [Peer Review File · Nature Communications]

Reviewers' Comments:

Reviewer #1:

Remarks to the Author:

This is an interesting article that should be of general interest to many readers. The authors present data showing a correlation between climatic conditions and the occurrence of rpoZ mutants in the Guertu plague focus of northwestern China.

I have no objections regarding the analyses that were done and think that the correlations shown are indeed real. I do, however, feel that the claims made with respect to the impact of the observed changes in frequency of the rpoZ mutants in the Gueta focus are very speculative and not supported by any data from the authors' studies. Although the author's contention that the extended phenotype changes exhibited through the effects of rpoZ-influenced differences in biofilm production on blockages of the flea's guts, the fleas' feeding behaviors, and the abilities of the blocked fleas to transmit *Y. pestis* efficiently, etc. are quite plausible and in general agreement with currently held concepts of transmission of *Y. pestis* by fleas, the study did not actually do blocking studies or transmission trials with infected fleas to show that the rpoZ mutants they observed actually affected the behavior of these strains in fleas relative to non-mutant strains or their ability to be transmitted by these insects. Although the notion of long-term survival of *Y. pestis* in soils remains contentious and unproven as a means of maintaining *Y. pestis* for long periods in nature, isn't it possible that the increase in frequency of the rpoZ mutants during certain climate conditions might also be related to their ability to survive within soils? I ask this because it seems that the observed changes in the frequency of rpoZ mutants during certain environmental conditions could have resulted in a number of effects other than just those involving flea-borne transmission. Can the author's address this issue? Also, if it is accepted that the rpoZ mutant strains increase in frequency following the described environmental conditions, what is the mechanism for this outcome? Increased survival in fleas? Increased transmission efficiency? Without accompanying data from laboratory experiments and transmission trials it is impossible to determine whether the effects are related to the infection of the flea vector, possible extrahost survival, etc.

The authors also mention the fact that overproduction of biofilm in rodent hosts negatively impacts *Y. pestis* virulence. The paper describing that work (Bobrove et al.) mentioned a different type of mutation in *Y. pestis*. Is there any reasons to suspect the rpoZ mutations would affect biofilm production in rodent hosts. Normally *Y. pestis* does not produce significant amounts of biofilm at the temperatures encountered in vertebrate hosts.

It seems strange that 133 strains of *Y. pestis* from locations outside the Guertu focus showed no rpoZ variation at all. It would seem that at least a few of the 133 non-Guertu strains examined would have shown similar mutations. Is it just inadequate sampling outside the Guertu focus or sampling at the wrong times (high precipitation periods, etc.)? Can the authors provide some explanation for this?

On page 7 of the manuscript the authors state that the ... "eight independently evolved rpoZ mutants observed appear to have been selected during low rodent and flea population densities..." I can agree with that but I think the rest of the sentence goes too far (...", presumably because low densities favor blockage-induced transmission."). As noted above, without doing actual transmission experiments with fleas infected with the strains containing the rpoZ mutation, it is probably too speculative to say this.

Reviewer #2:

Remarks to the Author:

This study examines the whether there is a selection pressure on *Y. pestis* modes of transmission in ground squirrels from Guertu. Using a genomic approach and ecological data the authors suggest that years of low precipitation, coupled with low host and vector abundance, explain the emergence of *rpoZ* mutants and the associated increased rate of biofilm production.

Changes in the pathogen-extended phenotype driven by climate is an attracting hypothesis. This implies that if there is flea blockage we should expect a relative increase in the density of fleas per host during those drier years as well as yersinia prevalence as a consequence of higher transmission. It would be helpful to elaborate more on the extended phenotype concept applied to this system and how this relates to flea density and yersinia prevalence.

Following this rationale, and what proposed in the study, I am surprised that there is no evidence of such mutants in the more recent years when precipitation and flea index are low. This should be clarified. How does changing the sampling area (different quadrats) have affected this pattern? and what was the frequency of these mutants in the host population?

What do those drier months mean for the bacterium? In other words, is a direct effect or the consequences of changes in host and vector abundance driven by climatic conditions and thus mode of pathogen transmission? A number of tradeoffs have been listed in the dynamics of plague and I don't think climate is necessarily a new one. While I can expect climatic conditions/seasonality to play a key role in host density and flea infestation I keep wondering if the ecological characteristics of this ecosystem are an important determinant, for example the ecology of the host species and the flea. In this respect, I would like the authors to elaborate more on this and why such mutations of the *rpoZ* gene have only be detected in this ecosystem.

I found difficult to re-conciliate figure 2 with figure 1-supplement. For example, in figure 1-sup. mutants occur when flea index and host abundance, as well as plague prevalence, are both low and high. Also some early mutants are no matched by host or vector data. In Fig. 2 fleas are presented as density and not as index. These patterns and data need to be clarified.

There are 8 mutants but only 7 are presented in the plots, am I missing something?

Reviewer #3:

Remarks to the Author:

I really enjoyed this ms, however there are many things that in my opinion are overlooked and results are over hyped. I will make general comments and then finer points pertaining to language clarification.

The authors have a really interesting system of plague in squirrels. Where they mention two possible transmission strategies. The use 78 genomes from 1967-2006. They claim to demonstrate that *rpoZ* mutants have an increased rate of biofilm production in vitro, and that the mutants tend to appear after the year with extremely dry autumns and winters. The claim that their results support the notion that the bacterium is constantly adapting.

First note: you have 78 genomes sampled over 40 years (while I understand that this is a very high number of sequences for the kind of system you have, unfortunately these are low numbers specially to make this sort of correlational analysis). Believe as a fellow ecologist it pains me to tell you sample size is not enough... The other problem is the sample distribution is not very balanced (I checked the year of isolation in one of the data tables). Making again the robustness of correlations harder to interpret, even with permutation analysis. This is something that needs to be shown upfront for

clarity!

While the conclusions are exciting, when looking at the methods, I'm not so confident in the methodology strength, as it is not quite clear exactly how these were calculated. Therefore, It's difficult for me to accept these at face value, what is assumed "under neutral substitution". The methods are poorly written and certain not reproducible! See specifics below for each section. Hence while I am trusting and tend to believe results, my role here is to evaluate and given the description and the absence of the code, I don't feel confident to take these results as they are.

One of the things you mention in passing is gene loss events occurred in only a few isolates, could actually have meaningful fitness implications—why didn't you develop this further?

Main text

Line 82 what do you mean by "The focus"

Line 106 what do you mean other 133 isolates? Is this other population of a similar system??

Line 124: "The ecological conditions in the ecosystem " this needs rephrasing

Methods

Host density-- Quadrat sampling for squirrel density? How do you correct for repeat counts? How do you deal with observational error? You need to refer to how you capture animals for the subsequent measures.

Flea Index – captured animals? In the Host density section, you don't mention captures. Also this estimation is assuming all squirrels have the same flea burden, what are the basis for that assumption?

Serop rate – how do you capture these animals you need to give a short even if one line description how this happens.

Bact isolation-- line 208: what do you mean " The vectors"??? What do you do with the squirrels do you release them or do you kill them to then "the organs of every tenth ground squirrel were fully mixed and injected intraperitoneally into a mouse " This methodology is not clear!!

Sequencing -- 54 high quality SNPs (28 non-synonymous, 10 synonymous and 16 intergenic) So dn/ds is not great.

What was the motivation to test out another genome of reference??

Mutation cluster—I would have like to see the code for the method.

Phylogeny reconstruction – PHYML with SNPs, that's fine. Would have liked to see the XML files for the BEAST2 analysis... Also why do BEAST if you had already done PHYML and the only thing you are using BEAST for is for the tree in figure 1. I'm not criticizing BEAST, just don't understand the justification.

Estimating local Temp and Preci—You didn't estimate anything, you collected the data. So you need to revise this title.

Statistics—What do you mean applied permutation tests to check if mutants co occurred with mutation? Do you mean bootstrapping and then regressions? This is not well explained, and needs further work. I would have liked to see the code for this analysis to make sure this is indeed bootstrapping and also what time of regression analysis was performed for appreciating "likelihood" of correlations.

While I appreciate that you seem confident that counting for Type I errors is not necessary, I am less so, I really would like to see details of the analysis to make sure... as associations between demographic dynamics (mutants emergence) and potentially related factors examined in post hoc fashions often ignore uncertainty in demographic reconstructions and the mechanisms.

Line 359: what do you. Mean by this " little precipitation would have a more substantial effect on

ecosystems”?

In the tree in Figure 1, I don’t “see” evidence for selection.

Nature Communications NCOMMS-18-31004-T**Responses to reviewers' comments and suggestions**

Reviewer #1

This is an interesting article that should be of general interest to many readers. The authors present data showing a correlation between climatic conditions and the occurrence of *rpoZ* mutants in the Guertu plague focus of northwestern China.

Comment 1

I have no objections regarding the analyses that were done and think that the correlations shown are indeed real. I do, however, feel that the claims made with respect to the impact of the observed changes in frequency of the *rpoZ* mutants in the Guertu focus are very speculative and not supported by any data from the authors' studies. Although the author's contention that the extended phenotype changes exhibited through the effects of *rpoZ*-influenced differences in biofilm production on blockages of the flea's guts, the fleas' feeding behaviors, and the abilities of the blocked fleas to transmit *Y. pestis* efficiently, etc. are quite plausible and in general agreement with currently held concepts of transmission of *Y. pestis* by fleas, the study did not actually do blocking studies or transmission trials with infected fleas to show that the *rpoZ* mutants they observed actually affected the behavior of these strains in fleas relative to non-mutant strains or their ability to be transmitted by these insects.

Author's response: We fully agree with the reviewer that experiments on the flea-borne transmission efficiency of *rpoZ* variants would further strengthen our conclusion. Unfortunately, no appropriate facilities with the required biosafety are available in the region of China where the fieldwork has been carried out. Such facilities do exist in Beijing, but for understandable biosafety reasons we are not allowed to ship isolates of pathogenic *Y. pestis* to Beijing. Hence, it is not feasible to conduct the requested flea-borne transmission experiments in the near future. We are, however, developing biosafety secure facilities in Yunnan – where such experiments can be done in due course. We have furthermore been in contact with Prof.

Joseph Hinnebusch (jhinnebusch@niaid.nih.gov) who is willing to do such future experiments in his laboratory in the US. Alternatively, Hinnebusch could join us in doing such experiments in the to-be-established new laboratory facilities in Yunnan. This, will for sure take quite a bit of time to finalize the necessary preparations, including addressing the necessary biosafety precautions.

Even in the absence of such an experiment we are convinced that our revised manuscript has substantially improved and is suitable for publication in *Nature Communications*, – not the least because we expanded the number of *Y. pestis* genome sequences outside of Guertu ecosystem which we screened for *rpoZ* variations from 133 to 368, and replaced the statistical analysis with one that is more robust, and which directly incorporates the dating uncertainty captured in the phylogeny. The novel methodology of the latter still confirms our conclusion that the rise of *rpoZ* variants is linked to the influence of climate on the ecosystem.

Comment 2

Although the notion of long-term survival of *Y. pestis* in soils remains contentious and unproven as a means of maintaining *Y. pestis* for long periods in nature, isn't it possible that the increase in frequency of the *rpoZ* mutants during certain climate conditions might also be related to their ability to survive within soils? I ask this because it seems that the observed changes in the frequency of *rpoZ* mutants during certain environmental conditions could have resulted in a number of effects other than just those involving flea-borne transmission. Can the author's address this issue? Also, if it is accepted that the *rpoZ* mutant strains increase in frequency following the described environmental conditions, what is the mechanism for this outcome? Increased survival in fleas? Increased transmission efficiency? Without accompanying data from laboratory experiments and transmission trials it is impossible to determine whether the effects are related to the infection of the flea vector, possible extrahost survival, etc.

The authors also mention the fact that overproduction of biofilm in rodent hosts negatively impacts *Y. pestis* virulence. The paper describing that work (Bobrove *et al.*) mentioned a different type of mutation in *Y. pestis*. Is there any reasons to suspect the *rpoZ* mutations would affect biofilm production in rodent hosts? Normally *Y. pestis* does not produce significant amounts of biofilm at the temperatures encountered in vertebrate hosts.

Author's response: The reviewer brings up the valid point that there are also other stages in the life cycle of *Y. pestis* that could be affected by climatic influences and variations that cause an overproduction of biofilm. Both the soil (a speculative compartment) and the vertebrate host compartment were not tested within this study, and we did not try to compare their relative importance compared to the flea compartment that was investigated here in-depth.

Regarding the soil compartment - in plague ecosystems where we had access to larger surveillance datasets, such as in Inner Mongolia, we looked at the role of climate on the various trophic levels of plague for two plague hosts – the Mongolian gerbil, and the Daurian ground squirrel¹. Neither of these species is fully representative of the Tien Shan long-tailed ground squirrel, but in both species we found no direct effects of climate (temperature or precipitation) on plague prevalence. Similarly, a study on plague in black-tailed prairie dogs in the USA emphasizes the indirect effect of climate on plague through the effects of temperature and humidity on flea development, which in turn influenced plague prevalence². Both studies therefore provide no support for a direct interaction between climate and the ability of the bacterium to survive in the soil, although it is clear that the bacterium itself has genes that are temperature-dependently expressed, and persistence of the bacterium in the soil, or in soil-associated amoeba remains a real possibility³.

Regarding the vertebrate host compartment – the general opinion for a long time has been that as biofilm formation is strongly reduced at 37°C *in vitro*, based on a study by Robert D. Perry et al.⁴. That opinion changed with studies from Young et al, 2012⁵ and Vadyvaloo et al, 2010⁶ who showed the reduction of biofilm formation that Perry et al. observed depended on the a-virulent plague strain that they used. The current view of the field is that biofilm formation might play an important role in keeping the innate immune system at bay in the first few hours after the transition from flea vector to rodent host⁶, and that virulent variants of *Y. pestis* actually upregulate PNAG – a major component of biofilms, rather than the downregulation of PNAG that Perry et al. observed.

We have adjusted the text to summarize the findings on the host compartment, and list citations on the former (there is not enough known yet to summarize how the soil compartment might interact with the dominant flea-rodent cycle). After corresponding with Joe Hinnebusch, we also slightly updated the relevant paragraph to be more precise about the trade-off. See line 64-79 “The two transmission modes ...” , and line 237-242 “Although there is strong experimental evidence..”

Comment 3

It seems strange that 133 strains of *Y. pestis* from locations outside the Guertu focus showed no *rpoZ* variation at all. It would seem that at least a few of the 133 non-Guertu strains examined would have shown similar mutations. Is it just inadequate sampling outside the Guertu focus or sampling at the wrong times (high precipitation periods, etc.)? Can the authors provide some explanation for this?

Author's response: Thank you for this comment. In the previous version of our submission, we did not detect SNPs in the *rpoZ* gene among the 133 non-Guertu *Y. pestis* strains. We have now obtained a more comprehensive dataset with 368 publicly available *Y. pestis* genomes (Dataset Table 6), and did find 6 strains with *rpoZ* variations outside of the Guertu ecosystem. These contained three new types of mutations in *rpoZ*, including one non-synonymous SNP and two indels (Extended Data Figure 4), as well as one indel identical to one in the Guertu strains. Most of the non-Guertu strains that carried *rpoZ* variants are isolated from the 0.PE group in the FSU region (Extended Data Figure 3), and one from a 2.MED group strain in Kyrgyzstan. This new finding indicates that *rpoZ* variations are more general for *Y. pestis* species than we earlier thought, based on the smaller sample of 133 non-Guertu strains. The manuscript in main text have been revised accordingly: please see Line 120-129 "Notably, the high frequency of polymorphisms..."

We did not further pursue the suggestion of the reviewer to look into the relationship between climate and *rpoZ* variants across the 133 strains, as we think that the expansion of the non-Guertu *Y. pestis* strains to 368 strains answers the reviewers' question. In addition, these strains cover such a large range of ecosystems and different rodent hosts, that we do not expect single climate thresholds to govern the behavior of these systems: the same level of precipitation may mean a drought in a montane meadow (accompanied by the appropriate amount of ecosystem stress), but be normal for a semi-arid desert^{1,7}.

Comment 4

On page 7 of the manuscript the authors state that the ... "eight independently evolved *rpoZ* mutants observed appear to have been selected during low rodent and flea population densities..." I can agree with that but I think the rest of the sentence goes too far (... "presumably

because low densities favor blockage-induced transmission."). As noted above, without doing actual transmission experiments with fleas infected with the strains containing the *rpoZ* mutation, it is probably too speculative to say this.

Author's response: Thank you for this comment. We have removed overly certain part of the sentence here. We still discuss this possibility (as per request of reviewer 2, comment 2) in Line 229-233 "Theoretical explorations of the conditions..", but in more careful terms.

Reviewer #2

This study examines the whether there is a selection pressure on *Y. pestis* modes of transmission in ground squirrels from Guertu. Using a genomic approach and ecological data the authors suggest that years of low precipitation, coupled with low host and vector abundance, explain the emergence of *rpoZ* mutants and the associated increased rate of biofilm production.

Comment 1

Changes in the pathogen-extended phenotype driven by climate is an attracting hypothesis. This implies that if there is flea blockage we should expect a relative increase in the density of fleas per host during those drier years as well as *Yersinia* prevalence as a consequence of higher transmission. It would be helpful to elaborate more on the extended phenotype concept applied to this system and how this relates to flea density and *Yersinia* prevalence.

Following this rationale, and what proposed in the study, I am surprised that there is no evidence of such mutants in the more recent years when precipitation and flea index are low. This should be clarified. How does changing the sampling area (different quadrats) have affected this pattern? And what was the frequency of these mutants in the host population?

Author's response: Thanks for these comments. We can indeed estimate the likelihood of sampling *rpoZ* variants post-1987 given the frequency with which *rpoZ* variants has been observed in the system, taking into account our suggestion that *rpoZ* variations would preferentially be sampled in years with low rodent and flea densities / suitable climate conditions (the thresholds reported) - see Figure below.

Following the above rationale, there are only a small number of years where both the flea index and the rodent density is known, and was low (i.e, years where the orange line and grey line have dots, and are "low" in Panel A in figure S1 (shown here)). These years would be 1991, 1996 & 1997. If we also look for years in which the climate thresholds we reported in the original version of the paper appear favorable for the evolution of *rpoZ* variants, we can putatively add the year 1999. In these 4 post-1987 years (1991, 1996, 1997 & 1999) we have two samples collected in 1991, no samples collected in 1996 and 1997, and one sample collected in 1999.

Given that the overall odds of finding *rpoZ* variants are fairly low (8/78 samples [10.3%] if

we disregard the climate-suitability of the year, and **8/39** samples [**20.5%**] if we are only looking for the odds of finding *rpoZ* variants in years with matching climate thresholds), the probability of not finding *rpoZ* variants after 1987 are not that extraordinary. During these 4 years with just three samples collected, and using the higher 20.5% chance of sampling *rpoZ* during suitable years, we get a probability of 50.2% of not detecting *rpoZ* variants post 1987, [calculating that probability as $(1 - 0.205)^3$].

As readers are likely to have the same question as the reviewer, we have now added a sub-figure to Figure 1b that shows the uneven sampling distribution over the years, and we give readers an approximate insight into the probabilities of sampling *rpoZ* variants in later years. And because the readers do not have access to the earlier version of the paper, with its emphasis on thresholds and low rodent and flea densities, we give the general case example in the text, rather than the more detailed example we provide here for the reviewer. See lines 130-137 “In the Guertu ecosystem, none of the *rpoZ*..” .

Comment 2

What do those drier months mean for the bacterium? In other words, is a direct effect or the consequences of changes in host and vector abundance driven by climatic conditions and thus mode of pathogen transmission? A number of tradeoffs have been listed in the dynamics of plague and I don't think climate is necessarily a new one. While I can expect climatic conditions/seasonality to play a key role in host density and flea infestation I keep wondering if the ecological characteristics of this ecosystem are an important determinant, for example the ecology of the host species and the flea. In this respect, I would like the authors to elaborate more on this and why such mutations of the *rpoZ* gene have only be detected in this ecosystem.

Author's response: During our research on the Guertu ecosystem, we explored how different trophic levels (weather, vegetation, rodent abundance, flea abundance) affect the bacterium, like has been done (including by us) for several other plague foci^{1,7-9}. We found increased springtime NDVI (plant growth) to be the most outstanding factor influencing the number of rodents, following by precipitation in autumn (potentially lengthening the season during which the ground squirrels can build up reserves) and snowfall in December (potentially providing isolation against the cold in winter burrows). However, we found the size of surveillance data of the Guertu focus to be too small to robustly support an analysis across all trophic levels, so we left this exploratory research out of the revised manuscript.

That said, we have summarized the literature on the topic in Lines 188-205 “How exactly colder and drier years affect the different trophic levels...”, to speculate on this particular ecosystem. As noted in Comment 3 in response to Reviewer 1, we found the *rpoZ* variation not to be unique to this ecosystem when screening a larger dataset of *Y. pestis* strains.

Comment 3

I found difficult to re-conciliate figure 2 with figure 1-supplement. For example, in figure 1-sup. mutants occur when flea index and host abundance, as well as plague prevalence, are both low and high. Also some early mutants are no matched by host or vector data. In Fig. 2 fleas are presented as density and not as index. These patterns and data need to be clarified.

Author’s response: We thank the reviewer for bringing up this point of confusion. As part of increasing the robustness of the statistical analysis and reducing over-interpretation of the results, we have moved away from using the earlier reported climate thresholds to predict what host and flea densities accompany those climate thresholds. As such, the confusion caused by presenting flea density (per hectare) in Figure 2 of the previous version, and flea index (per rodent) in Supplementary Figure 1 is now no longer relevant.

Looking at Figure S1 indeed suggests that *rpoZ* variants can occur both at low and high flea index, host abundance and plague prevalence, seemingly contradicting the findings presented in the paper. The confusion is caused by presenting the raw surveillance and sampling data in Figure S1 without sufficient explanation that selection for the persistence and possible expansion of *rpoZ* variants has happened somewhere in the period between the emergence of these variants (previously not indicated in the Figure) and their sampling (indicated by a red dot). Solely presenting the *rpoZ* variants as red dots gives the impression that one can just look at a vertical slice of the data in the year of sampling. We have now added lines leading up to a red dot in Figure S1, indicating the period during which the *rpoZ* variants were detected up to their moment of sampling (as estimated by the median divergence date of the *rpoZ* variants in the phylogenetic analysis). This should make clear that it is not necessarily the number of rodents or fleas in the year of sampling that is relevant for the divergence of *rpoZ* variants. We have updated Figure S1 (see Comment 1) and the legend accordingly.

The reviewer also correctly notices that the available surveillance data does not go back as far as the genetic sampling and storing of plague strains. The strains were well stored since its isolation as early as 1960s, which provided us information of genetic diversity. However, the surveillance information during that period were recorded by hand writing or mimeograph, and some of them had been damaged and the information cannot be accurately recovered. There is unfortunately no additional surveillance data available. We have now explicitly commented on that in the legend of Figure S1, and in the accompanying text in the Extended Data document.

Comment 4

There are 8 mutants but only 7 are presented in the plots, am I missing something?

Author's response: Thanks for pointing this out. In our initial figure, the dots indicated the *years* in which *rpoZ* variants were sampled, and so the year 1983 in which two *rpoZ* variants were sampled was displayed only as a single dot. We have now changed the legend and content of Supplementary Figure 1, such that all 8 variants are plotted.

Reviewer #3

I really enjoyed this ms, however there are many things that in my opinion are overlooked and results are over hyped. I will make general comments and then finer points pertaining to language clarification.

Comment 1

The authors have a really interesting system of plague in squirrels. Where they mention two possible transmission strategies. The use 78 genomes from 1967-2006. They claim to demonstrate that *rpoZ* mutants have an increased rate of biofilm production in vitro, and that the mutants tend to appear after the year with extremely dry autumns and winters. The claim that their results support the notion that the bacterium is constantly adapting.

First note: you have 78 genomes sampled over 40 years (while I understand that this is a very high number of sequences for the kind of system you have, unfortunately these are low numbers specially to make this sort of correlational analysis). Believe as a fellow ecologist it

pains me to tell you sample size is not enough... The other problem is the sample distribution is not very balanced (I checked the year of isolation in one of the data tables). Making again the robustness of correlations harder to interpret, even with permutation analysis. This is something that needs to be shown upfront for clarity!

Author's response: We agree with the reviewer and have added a sub-figure with the sampling distribution to Figure 1, and in the text describe the limits of the sample size and its distribution in Lines 130-137 "In the Guertu ecosystem, none of the *rpoZ* variations...".

We furthermore share the reviewer's concern regarding the robustness of the results and scaled back part of the extrapolation of the results (where we use the thresholds to predict what the rodent and flea densities would have been during *rpoZ* years). Most notably, we changed the analysis of the correlation between climate and the divergence of *rpoZ* variants such that we no longer depend on the assumption that the variants appeared somewhere during the year prior to sampling – rather, we use the (raw intermediate) phylogenetic trees generated by the MCMC to inform us of the distribution of dates when each of the *rpoZ* variations could have been generated. As we do not know when during the period between their divergence and sampling *rpoZ* variants were experiencing circumstances that (likely) boosted their prevalence, we look at the average climate during the whole interval between generation and sampling of the *rpoZ* variant (as well as a number of months prior to the generation of the variation, to account for trophic cascade effects of climate affecting rodents and fleas, which then in turn could have created the circumstances that favored *rpoZ* variants). As said, the benefits of the new approach is that we drop an assumption on **when** the mutations were under positive selection pressure, and we incorporate the uncertainty captured in the process of making the phylogenetic tree (we use thousands of the MCMC-generated phylogenetic trees). We also take the simpler approach of looking at the average monthly temperature and precipitation levels, rather than splitting up the analysis over specific months. See also Figure 2 and lines 148-186 "A general caveat when comparing biofilm formation..." and lines 438-59 " As described in the main text, we used permutation..." in the text, and Comment 16 and 17 for a further discussion on the new methodology, and our testing for Type I errors.

Comment 2

While the conclusions are exciting, when looking at the methods, I'm not so confident in the methodology strength, as it is not quite clear exactly how these were calculated. Therefore, it is

difficult for me to accept these at face value, what is assumed "under neutral substitution". The methods are poorly written and certain not reproducible! See specifics below for each section. Hence while I am trusting and tend to believe results, my role here is to evaluate and given the description and the absence of the code, I don't feel confident to take these results as they are.

Author's response: Our apologies for causing concerns. We cleaned up the description of the computational methods. For the clustering method, we rewrote our description of the algorithm that we used and simplified our workflow (and description thereof) by which we avoided testing subsets of clusters that were already found to be significant. See Lines 374-393 "We searched for any cluster of variations...". The statistics for the climate analysis changed substantially in response to reviewer comments and the new version is discussed at Comment 16 and 17, and available in the manuscript at Lines 148-186. " A general caveat when comparing biofilm formation..." and lines 438-485 " As described in the main text, we used permutation...". The earlier version of the manuscript did include a link to the code repository in the supplementary materials, which contained data files, source code, and source code comments in that could have alleviated some of the reviewer's concerns, but was evidently placed in a spot that was easy to overlook. We have now opted to mention the code repository more prominently throughout the text (the link is still at the end of the manuscript). In updating the methods, we have also updated the code repository (the new repository is available at <https://doi.org/10.5281/zenodo.3279956>), and in addition included an XML file to repeat the BEAST analysis, and a README.txt with how to run the code in the code repository.

Comment 3

One of the things you mention in passing is gene loss events occurred in only a few isolates, could actually have meaningful fitness implications-why didn't you develop this further?

Author's response: Thank you for this comment. The gene loss events focused on two genomic islands, which are known to be unstable in *Y. pestis*¹⁰⁻¹². We cannot confirm that the loss event occurred in natural environments or during laboratory passage, therefore, to avoid possible misleading results, we excluded them in further analysis.

Main text

Comment 4

Line 82 what do you mean by "The focus"

Author's response: "The focus" refers to "The natural plague focus", we have revised the sentence, please see Line 90-91 " We investigated the temporal dynamics of *Y. pestis*... "

Comment 5

Line 106 what do you mean other 133 isolates? Is this other population of of a similar system??

Author's response: The "other 133 isolates" are global isolates of *Y. pestis* representing the diversity of whole species in our previous work. To further include a more global genetic diversity of *Y. pestis* species, in the revised version, we now expanded the 133 isolates to 368 publicly available *Y. pestis* genomes and searched for mutations in all of these within the *rpoZ* gene. Please see Line 120-129 "Notably, the high frequency of polymorphisms in the..." for detail.

Comment 6

Line 124: "The ecological conditions in the ecosystem". This needs rephrasing

Author's response: The manuscript had been extensive revised and this sentence had been removed from the new version.

Methods

Comment 7

Host density-- Quadrat sampling for squirrel density? How do you correct for repeat counts?
How do you deal with observational error?

Author's response: The surveillance of plague in Guertu was performed according to standard protocol in National Scheme of Plague Surveillance of China. As the Guertu natural plague focus is located at a remote mountainous region, the only reasonable method for determining its host density during 1970s is quadrat sampling with ocular estimation, and then same protocol was adopted in the following years. In each quadrat, the number of ground squirrels was counted by telescope one hour after sunrise, the peak time for activity of local ground squirrel, which could reduce the possibility of under estimation of the population size. The count lasted for two hours and was repeated for two consecutive days. The highest count recorded within the two days was used to calculate host density by dividing the number of ground squirrels by the

quadrat area. Host density for the entire Guertu region was calculated as the average density between quadrats to reduce the observation error for each quadrat. While the system error can still present in calculating the host density by using above methods, the fluctuation pattern of the host density in this plague focus is comparable among different years as identical surveillance protocol was applied for each year.

In the current version of the manuscript, where we are more careful in how far we take our results, the observational data is no longer directly used in the statistical analysis, but only presented as supporting information.

Comment 8

You need to refer to how you capture animals for the subsequent measures.

Author's response: Thank you for this comment. We have added a section named "Capture of host animals" in the Method section. Please see Lines 281-286.

Comment 9

Flea Index - captured animals? In the Host density section, you don't mention captures. Also this estimation is assuming all squirrels have the same flea burden, what are the basis for that assumption?

Serop rate - how do you capture these animals you need to give a short even if one line description how this happens.

Author's response: We have added the description on captures of animals. The number of fleas on each captured animal was counted and recorded independently. As in this study we used the average value of flea index for a year, we do not need to assume that all hosts (ground squirrels) have the same flea burden.

Comment 10

Bact isolation-- line 208: what do you mean "The vectors"??? What do you do with the squirrels do you release them or do you kill them to then "the organs of every tenth ground squirrel were fully mixed and injected intraperitoneally into a mouse " This methodology is not clear!!

Author's response: The "vectors" denotes fleas that were collected from the captured animal. The ground squirrels were executed in the field and after carried back to the laboratory, the

serum and organs were sampled from each animal for further work. We have revised the Method section to clarify the process of the plague surveillance work.

Comment 11

Sequencing -- 54 high quality SNPs (28 non-synonymous, 10 synonymous and 16 intergenic)
So dn/ds is not great.

Author's response: The ratio between nonsynSNP and synSNP of Guertu isolates is moderate high compared to the values of the species as a whole (see below table). This is in concordance with the estimations of neutral evolution, i.e., a recently evolved population would reveal a higher dN/dS value as some slightly detrimental mutations have not had sufficient time to be removed by purifying selection.

Number of SNPs			Ratio Nonsyn/Syn	of Reference
Nonsynonymous	Synonymous	Intergenic		
901	409	369	2.2	Morelli et al, 2010, Nat Genet ¹³
1295	572	366	2.3	Cui et al, 2013, PNAS ¹⁴
28	10	16	2.8	This study

Comment 12

What was the motivation to test out another genome of reference??

Author's response: The commonly used reference genome of *Y. pestis*, CO92, belongs to 1.ORI phylogroup, which is genetically a quite distant relative of the Guertu strains (0.ANT1, Extended Data Figure 3), and the Guertu samples might have harbored rearrangements or large variable genomic fragments that are not present in CO92. Reassembling the Guertu genomes against a reference genome that belonged to the same phylogroup as the Guertu samples somewhat reduces the risk of falsely calling SNPs at the edges of rearrangements of variable

genomic fragments. In retrospect, this appears not to have been necessary, and we could have used the CO92 reference genome as well.

Comment 13

Mutation cluster-I would have like to see the code for the method.

Author's response: The full source code and files needed to replicate the computational results, including the mutation clustering are available at <https://doi.org/10.5281/zenodo.3279956>. This repository was referred to in the supplemental materials of the previous version of the manuscript as well (at <https://zenodo.org/record/1321605>), but we have now made sure to mention the code repository more prominently throughout the text.

Comment 14

Phylogeny reconstruction - PHYML with SNPs, that's fine. Would have liked to see the XML files for the BEAST2 analysis... Also why do BEAST if you had already done PHYML and the only thing you are using BEAST for is for the tree in figure 1. I'm not criticizing BEAST, just don't understand the justification.

Author's response: Thank you for this comment. We chose to use PHYML to generate the whole-species tree of more than 200 genomes (Extended Data Fig. 3, in current version in total 446 genomes were used to build the tree), which saved computational time and resources compared to BEAST. On the smaller subset of samples presented in Figure 1, we used BEAST, and then proceeded to use the estimated divergence times of the nodes from BEAST in our statistical analysis in Figure 2 (see also comment 17). It is not uncommon to use ML methods to build the whole species tree, and then using BEAST to estimate divergence times of nodes; see for example Simon Rasmussen et al, Cell, 2015¹⁵. The XML file has been added to the new zenodo code repository listed in the manuscript: see here: <https://doi.org/10.5281/zenodo.3279956>

Comment 15

Estimating local Temp and Precipitation - you didn't estimate anything, you collected the data. So you need to revise this title.

Author's response: Thank you for this comment. We earlier experimented with a more complex estimation of local climate, but then simplified our analysis (as it had no impact on the

results), and the old title slipped through. We revised the title to now read “Collection of local temperature and precipitation data”.

Comment 16

Statistics-What do you mean applied permutation tests to check if mutants co occurred with mutation? Do you mean bootstrapping and then regressions? This is not well explained, and needs further work. I would have liked to see the code for this analysis to make sure this is indeed bootstrapping and also what time of regression analysis was performed for appreciating "likelihood" of correlations.

Author's response: We have clarified the text to be more specific with respect to details. The original full source code and files needed to replicate the computational results, including the permutation testing were (and still are) available at <https://zenodo.org/record/1321605>. This repository was linked to in the previous version of the manuscript in the supplemental materials, but we have now made sure to mention the code repository more prominently throughout the text (link is still at the end). Since we revised our manuscript to more tightly integrate the uncertainty captured in the phylogenetic tree MCMC and the generation time of the *rpoZ* variants, we rewrote the related section, and created a new code repository at <https://doi.org/10.5281/zenodo.3279956>. See Lines 148-186. " A general caveat when comparing biofilm formation..." and lines 438-485 "As described in the main text, we used permutation..."

Comment 17

While I appreciate that you seem confident that counting for Type I errors is not necessary, I am less so, I really would like to see details of the analysis to make sure... as associations between demographic dynamics (mutants emergence) and potentially related factors examined in post hoc fashions often ignore uncertainty in demographic reconstructions and the mechanisms.

Author's response: We share the reviewers' concern on Type I errors. As partially answered in Comment 1, we changed our analysis on climate and *rpoZ* variants to consider the climate over the time period covered by the whole branch of the tree during which an *rpoZ* variant evolved (plus an additional time period before – see new method section at lines 438-485 "As described in the main text, we used permutation..."), rather than to assume that the *rpoZ* variant must have arisen somewhere in the year prior to sampling.

The benefit of the new analysis is that it allows us to make use of the intermediate

phylogenetic trees generated by BEAST during its MCMC run, and thus capture a bit more of the demographic reconstruction uncertainty in the system directly in the analysis, and even more importantly – this approach more accurately reflects the potential climatic conditions that were present during the time the *rpoZ* variants evolved.

In contrast to our earlier approach, we no longer split up the year into separate months or clusters of months, but have kept the comparison simpler in that regard by comparing the average monthly temperature or average monthly precipitation levels in the phylogenetic-tree informed time period during which the *rpoZ* variants might have evolved. That leaves us with 8 tests – the climate was (warmer, cooler, drier, wetter, warmer & wetter, warmer & drier, colder & wetter, colder & drier) during the time *rpoZ* variants appeared compared to the time period preceding the other *Y. pestis* samples.

Two of these tests were significant at the p -value <0.05 level (one at 0.03 and one at 0.0092), but prior to correcting for multiple testing. To correct for the multiple testing, we applied another layer of permutation testing – picking 8 random samples (without resampling) from the 78 samples as our "samples of interest" (like the *rpoZ* variants are samples of interest in our study), and performing permutation tests to compare how often in our system with its sampling biases and uncertainty in divergence dates, we get significant results with a p -value of 0.0092 or less in at least one of the 8 climate tests.

Our main results (new added Figure 2 in main text, see below) now includes both the significant climate association found (colder & drier average monthly climate), and the results of our multiple-testing analysis, thus answering the reviewers' concern and putting more weight on the importance of robustness testing of your results.

Caption: Panel A: average climate conditions during the periods in which the *rpoZ* variants were generated (as per the branch lengths of the phylogenetic tree, plus a year prior to account for

cascading effects of climate on hosts and vectors to plague), compared to the climate conditions for *rpoZ* references. **Panel B:** correcting the observed p-value of < 0.0092 for the pattern of low average monthly precipitation and temperature observed in Panel A for multiple testing of 8 related climate hypothesis, using permutation testing (orange line) as opposed to the Bonferroni correction (which assumes unrelated tests). The corrected p-value is $p < 0.046$.

We realize that with our change in methodology some details have disappeared in the revised version – no longer do we try to pin down the most relevant months, or predict the host and vector conditions in the paper based on the climate conditions, but in return we think we substantially increased the robustness of our results, also by directly integrating information about uncertainty from the phylogenetic analysis into these results. We thank the reviewer for pushing us into this direction and as a result of the robustness testing have become a bit more conservative in how far we were willing to push the data interpretation. The results are in our opinion improved and a more accurate use of the data than in the original submission. We are grateful for these comments that have improved our paper and believe that the revised text now makes the strength and robustness of our approach clear.

Comment 18

Line 359: what do you mean by this "little precipitation would have a more substantial effect on ecosystems"?

Author's response: The original sentence meant to argue that multi-month periods of drought would have a disproportional impact on an ecosystem compared to single-month droughts (e.g. more heavily depleting water reserves in the ground and in the vegetation). Multi-month droughts were therefore expected to exert a more substantial selection pressure on the actors of the ecosystem. As the methods for this part of the manuscript have been extensively revised, this sentence is no longer relevant, and it has been removed from the new version of manuscript.

Comment 19

In the tree in Figure 1, I don't "see" evidence for selection.

Author's response: the strongest signal for selection that we found was in the *rpoZ* gene. The evidence for selection one can “see” in the phylogenetic tree in Figure 1 is that in an unlikely 8 out of 78 samples, a very small gene within the *Y. pestis* genome contained variations (the red dots) - a signal of convergent evolution and consequently this gene is under selection pressure. Besides, after including more genomes in this round of revision, we found more *rpoZ* variants in different *Y. pestis* lineages, which further support this gene is under selection.

The reviewer might has a point that because none of the 8 *rpoZ* variants were observed in more than one sample, we do not “see” evidence in the tree in the form of selective sweeps, where the *rpoZ* variant propagates through the population to become the dominant form.

The same is true for the other candidate selected-for mutations listed in Table 1. Only in one case do we find one of these mutations to have been resampled frequently over the years (the one listed as Intergenic; YPO0535–YPO0536, but with a high probability of not being different from a neutral substitution model at $p = 0.013$). Most other mutations listed in Table 1 are unique to their sample, with some occurring in 1 or 2 other samples. As such, the phylogenetic tree would not show any signs of selective sweeps happening. Now, for what reason the candidate selected-for variations do not propagate is hard to tell; we would speculate that in the case of the *rpoZ* variations, the circumstances under which they are selected for are short-lived – due to short term periodically climatic fluctuations. It has been found in *Salmonella enterica* serova Paratyphi A, the historical genetic changes that under Darwinian selection are continuously removed by following purifying selection¹⁶. The phenomena was named transient Darwinian selection, and was inferred related with environmental changes. Here what we observed in Guertu ecosystem could be another case of transient Darwinian selection in the bacterial world.

References

- 1 Xu, L. *et al.* The trophic responses of two different rodent-vector-plague systems to climate change. *Proceedings Biological Sciences* **282**,

-
- 20141846 (2015).
- 2 Savage, L. T., Reich, R. M., Hartley, L. M., Stapp, P. & Antolin, M. F. Climate, soils, and connectivity predict plague epizootics in black-tailed prairie dogs (*Cynomys ludovicianus*). *Ecological Applications* **21**, 2933-2943 (2011).
 - 3 Benavides-Montaña, J. A. & Vadyvaloo, V. *Yersinia pestis* resists predation by *Acanthamoeba castellanii* and exhibits prolonged intracellular survival. *Appl. Environ. Microbiol.* **83**, e00593-00517 (2017).
 - 4 Perry, R. D. *et al.* Temperature regulation of the hemin storage (Hms+) phenotype of *Yersinia pestis* is posttranscriptional. *J Bacteriol* **186**, 1638-1647 (2004).
 - 5 Yoong, P., Cywes-Bentley, C. & Pier, G. B. Poly-N-acetylglucosamine expression by wild-type *Yersinia pestis* is maximal at mammalian, not flea, temperatures. *MBio* **3**, e00217-00212, doi:10.1128/mBio.00217-12 (2012).
 - 6 Vadyvaloo, V., Jarrett, C., Sturdevant, D. E., Sebbane, F. & Hinnebusch, B. J. Transit through the flea vector induces a pretransmission innate immunity resistance phenotype in *Yersinia pestis*. *PLoS Pathog* **6**, e1000783, doi:10.1371/journal.ppat.1000783 (2010).
 - 7 Collinge, S. K. *et al.* Testing the Generality of a Trophic-cascade Model for Plague. *Ecohealth* **2**, 102-112 (2005).
 - 8 Stenseth, N. C. *et al.* Plague dynamics are driven by climate variation. *Proc Natl Acad Sci U S A* **103**, 13110-13115, doi:10.1073/pnas.0602447103 (2006).
 - 9 Kausrud, K. L. *et al.* Climatically driven synchrony of gerbil populations allows large-scale plague outbreaks. *Proc Biol Sci* **274**, 1963-1969, doi:10.1098/rspb.2007.0568 (2007).
 - 10 Brubaker, R. R. Mutation rate to nonpigmentation in *Pasteurella pestis*. *J Bacteriol* **98**, 1404-1406 (1969).
 - 11 Parkhill, J. *et al.* Genome sequence of *Yersinia pestis*, the causative agent of plague. *Nature* **413**, 523-527, doi:10.1038/35097083 (2001).

-
- 12 Tong, Z. *et al.* Genetic variations in the *pgm* locus among natural isolates of *Yersinia pestis*. *The Journal of general and applied microbiology* **51**, 11-19 (2005).
 - 13 Morelli, G. *et al.* *Yersinia pestis* genome sequencing identifies patterns of global phylogenetic diversity. *Nat Genet* **42**, 1140-1143, doi:ng.705 [pii] 10.1038/ng.705 (2010).
 - 14 Cui, Y. *et al.* Historical variations in mutation rate in an epidemic pathogen, *Yersinia pestis*. *Proc Natl Acad Sci U S A* **110**, 577-582, doi:10.1073/pnas.1205750110 (2013).
 - 15 Rasmussen, S. *et al.* Early divergent strains of *Yersinia pestis* in Eurasia 5,000 years ago. *Cell* **163**, 571-582, doi:10.1016/j.cell.2015.10.009 (2015).
 - 16 Zhou, Z. *et al.* Transient Darwinian selection in *Salmonella enterica* serovar Paratyphi A during 450 years of global spread of enteric fever. *Proc Natl Acad Sci U S A* **111**, 12199-12204, doi:10.1073/pnas.1411012111 (2014).

Reviewers' Comments:

Reviewer #1:

Remarks to the Author:

The reviewer thanks the authors for addressing his concerns in a thorough manner. Although I find the work very interesting, I still believe the conclusions are highly speculative and would feel much more confident in the authors assertions if they were backed up by other experimental work with flea transmission, etc.

Other points:

The opening sentence in the abstract and lines 58-62 are not entirely correct. Although no one doubts that biofilm production increases the overall efficiency of flea-borne transmission of *Y. pestis* when the foreguts of these insects become blocked by a biofilm-containing mass, and other evidence indicates transmission can occur as a result of partial blockage with biofilm during the early phase period, the work of Vetter et al. (2010) clearly demonstrates that biofilm is not required for early phase transmission and the rates of transmission of biofilm-deficient mutants during the early phase period are similar to those observed for biofilm producing strains. Please correct this misleading statement.

Lines 80-82: This sentence seems confusing. Should the phrase "among bacteria be specific feature to *Y. pestis*" be changed to read "and among bacteria be a feature specific to *Y. pestis*"?

The authors do not mention how many times the strains were passaged prior to being sent to the Chinese Medical Bacteria Center of Management and Preservation for storage as a freeze dried powder. Considering 7 of the 78 strains had lost the pathogenicity island or *pgm* locus, one might worry that the strains were high passage and that this affect selection for the different *rpoZ* variants. Was this considered?

Reviewer #2:

Remarks to the Author:

I have reviewed a prior version of this manuscript and I appreciate the effort of the authors to provide additional genomics data and new/revised results in support of their conclusions. I also think that the concerns raised in the previous version were addressed thoroughly. The work reads much better and the general conclusions hold more clearly. I have few comments on the current version.

The two modes of transmission are suggested to be in a trade-off between each other although it 'is more complex than the straightforward trade-off' (line 68-69). Based on these complexities is it really a trade-off? A clarification will be helpful.

Where are the *zpoZ* references coming from, and why/based on what are they considered a reference? This is still unclear to me.

Line 172. 'The other 7 climate hypotheses were borderline significant or non-significant'. Based on some of these you can't completely dismiss possible alternative/complementary effects. It will be useful to clarify and comment on what these alternative borderline significant climatic hypotheses are.

In the Methods there is information that was not used for the work that can actually be removed (e.g. seropositivity rate).

Reviewer #3:

Remarks to the Author:

I am very happy with the changes the authors made.

Nature Communications NCOMMS-18-31004A**Responses to reviewers' comments and suggestions**

Reviewer #1**Comment 1**

The reviewer thanks the authors for addressing his concerns in a thorough manner. Although I find the work very interesting, I still believe the conclusions are highly speculative and would feel much more confident in the authors assertions if they were backed up by other experimental work with flea transmission, etc.

Author's response: We agree with the reviewer that experiments on the flea-borne transmission efficiency of *rpoZ* variants would further strengthen our conclusion. However, as explained in our previous feedback regarding this suggestion, there are no appropriate facilities with the required biosafety level available in the larger region around the fieldwork location and away from population centers. We are currently working on developing appropriate biosafety secure facilities for performing such experiments in Yunnan, and are in contact with Prof. Joseph Hinnebusch (jhinnebusch@niaid.nih.gov) to discuss cooperation for these experiments. However, it will for sure take quite some time to make the necessary preparations for such experiments, and hence it is not feasible for us to conduct the requested flea-borne transmission experiments in the near future.

Comment 2

The opening sentence in the abstract and lines 58-62 are not entirely correct. Although no one doubts that biofilm production increases the overall efficiency of flea-borne transmission of *Y. pestis* when the foreguts of these insects become blocked by a biofilm-containing mass, and other evidence indicates transmission can occur as a result of partial blockage with biofilm during the early phase period, the work of Vetter et al. (2010) clearly demonstrates that biofilm is not required for early phase transmission and the rates of transmission of biofilm-deficient mutants during the early phase period are similar to those observed for

biofilm producing strains. Please correct this misleading statement.

Author's response: Many thanks for pointing this out. To avoid possible confusion, we revised the sentences in the Abstract and Main text. Now the sentences read: Lines 33-34 "...alternatively, during a brief period directly after feeding on a bacteremic host." in the Abstract, and Lines 56-60 "... The second transmission mode is known as early-phase transmission³. While the existence of the transmission route is well-documented, the mechanism of transmission is as of yet unknown. Early-phase transmission in fleas fed on mice blood has been shown to transmit in similar efficiency in biofilm-deficient *Y. pestis* strains as in biofilm producing strains⁵." in the Main text. We also introduce a new citation (Reference 3) to enhance this statement.

Comment 3

Lines 80-82: This sentence seems confusing. Should the phrase "among bacteria be specific feature to *Y. pestis*" be changed to read "and among bacteria be a feature specific to *Y. pestis*"?

Author's response: Thank you for pointing this out. We revised the sentence to read "...The rate of biofilm production is a genetically determined trait of *Y. pestis*, and alters the feeding behavior of the flea by blocking its proventriculus. Among bacteria, the ability to modulate the fleas' feeding behavior appears to be a feature specific to *Y. pestis*.". Please see Lines 86-88.

Comment 4

The authors do not mention how many times the strains were passaged prior to being sent to the Chinese Medical Bacteria Center of Management and Preservation for storage as a freeze dried powder. Considering 7 of the 78 strains had lost the pathogenicity island or *pgm* locus, one might worry that the strains were high passage and that this affect selection for the different *rpoZ* variants. Was this considered?

Author's response: We checked the original documents to find the passage record of the strains. Most of the strains experienced less than five passages and then were kept as freeze-dried powder. Two strains, 42005 and 42010, had been passaged for more than 30 times before being sent to the culture center. Indeed, GI03 was lost in both strains. However, the loss of *pgm* locus and GI03 in other strains seemed random, and the occurrence of *rpoZ* variants is also unrelated with the number of passages. Besides, we observed that one *rpoZ* variant occurred in a lineage of 0.PE strains (Supplementary Figure 3). It seems this *rpoZ* variation had been fixed

in the 0.PE phylogroup, which is unlikely to be explained by selection during laboratory passage.

Indirect evidence (and therefore not included in the paper) that passaging does not cause the appearance of *rpoZ* variants is evident by looking at one of the most commonly used live attenuated plague vaccines, namely the EV76 lineage of *Y. pestis*. EV76 was derived from a strain isolated from Madagascar and after 76 serial passages over 6 years lost its *pgm* locus and became strongly attenuated to humans¹. It has since been used as a human attenuated plague vaccine in plague endemic areas globally^{2,3}. Up until now, six genotype variants of the original EV76 have been identified that formed during passaging in laboratories and in vaccine production⁴. We find no *rpoZ* variants in the four EV76 genotypes for which sequence data is available, supporting the idea that the *rpoZ* gene is at least fairly conserved during laboratory passage.

Reviewer #2

I have reviewed a prior version of this manuscript and I appreciate the effort of the authors to provide additional genomics data and new/revised results in support of their conclusions. I also think that the concerns raised in the previous version were addressed thoroughly. The work reads much better and the general conclusions hold more clearly.

Author's response: We appreciate this positive comment.

Comment 1

I have few comments on the current version. The two modes of transmission are suggested to be in a trade-off between each other although it 'is more complex than the straightforward trade-off' (line 68-69). Based on these complexities is it really a trade-off? A clarification will be helpful.

Author's response: Thank you for this comment. We revised the corresponding sentences to clarify the “trade-off” between two transmission modes. Now the sentences read as “. . . This trade-off between blockage-induced transmission and early-phase transmission occurs in the domain of normal-to-high levels of biofilm production. In the domain of normal-to-reduced levels of biofilm production, the relationship between blockage-induced transmission and early-phase transmission becomes more complex, depending on the host blood source. For mice, normal levels of biofilm formation, or even the complete absence of biofilm formation

does not seem to negatively impact early-phase transmission, while it does negatively impact blockage-induced transmission. In contrast, in rats and guinea pigs, low or absent levels of biofilm formation does negatively impact early-phase transmission, as some degree of biofilm formation appears to be involved in an interaction between host blood and *Y. pestis* that drastically boosts the efficiency of early-phase transmission. In addition, at reduced levels of biofilm production, more fleas will be at a stage of partial, rather than complete blockage of the proventriculus. As partially blocked fleas can still hydrate themselves, yet also spread plague, reduced levels of biofilm production increases the bacterium's ability to survive in hibernating fleas, and extends the lifespan of infected fleas - both factors improving the persistence of plague in the ecosystem.". Please see Lines 67-81.

Comment 2

Where are the *zpoZ* references coming from, and why/based on what are they considered a reference? This is still unclear to me.

Author's response: As we described in Lines 165-166, the *rpoZ* references indicate "strains that have an *rpoZ* sequence identical to the CO92 reference strain". To avoid possible confusion, we also added sentences in the Method section, which read "Among all 446 *Y. pestis* genomes analyzed in this study, the sequence of *rpoZ* gene is fully identical in 403 genomes. Here we defined the strains that carried major allele type of the *rpoZ* as "*rpoZ* reference" (identical to the CO92 reference strains). And the other 43 genomes that carried minor allele types of *rpoZ* were defined as "*rpoZ* variants"". Please see Lines 455-458.

Comment 3

Line 172. 'The other 7 climate hypotheses were borderline significant or non-significant'. Based on some of these you can't completely dismiss possible alternative/complementary effects. It will be useful to clarify and comment on what these alternative borderline significant climatic hypotheses are.

Author's response: Thank you for this comment. We now discuss the two borderline-significant climate hypotheses in the text as well. Together with the strongest climate hypothesis they cover all possible scenarios that include a "low precipitation" component into the climate hypothesis, suggesting that the "low precipitation" component is the more important factor, and the role of temperature is less certain. We revised the corresponding sentences, which now read as follows: "...The other 7 climate hypotheses were

borderline significant or non-significant, with the two most significant of these (at p-values of $p < 0.0396$ and $p < 0.0304$) indicating that *rpoZ* variants arose during climate periods that were drier, and warmer & drier, respectively. While neither of these hypotheses would remain significant after correcting for multiple testing, they may indicate that a lack of precipitation is the more important factor in the rise of *rpoZ* variants, and that the role of temperature is less certain.

Applying a Bonferroni correction to correct for multiple testing of the different climate hypotheses would be overly stringent here, as our 8 hypotheses are correlated with each other...” Please see Lines 199-207.

Comment 4

In the Methods there is information that was not used for the work that can actually be removed (e.g. seropositivity rate).

Author’s response: Thank you for pointing this out. The section on Seropositivity rate has now been removed from the manuscript.

Reviewer #3

I am very happy with the changes the authors made.

Author’s response: We appreciate this very positive comment.

References

- 1 Girard, G. & Robic, J. Current Status of the Plague in Madagascar and Vaccinal Prophylaxis with the Aid of the EV Virus-Vaccine. *Bull. Soc. Path. exot.* **35**, 43-49 (1942).
- 2 Gallut, J. & Girard, G. [Action of chlorpromazine on experimental poisoning and infection of the mouse by *Pasteurella pestis* (EV vaccinal strain)]. *Ann Inst Pasteur (Paris)* **100**, 672-676 (1961).
- 3 Girard, G. Immunity in plague. Acquisitions supplied by 30 years of work on the "Pasteurella pestis EV" (Girard and Robic) strain. *Biol Med (Paris)* **52**, 631-731 (1963).
- 4 Cui, Y. *et al.* Genetic variations of live attenuated plague vaccine strains (*Yersinia*

pestis EV76 lineage) during laboratory passages in different countries. *Infect Genet Evol* **26**, 172-179, doi:10.1016/j.meegid.2014.05.023 (2014).